# Dual Inhibition of HIF-1α and HIF-2α as a Promising Treatment for VHL-Associated Hemangioblastomas: A Pilot Study Using Patient-Derived Primary Cell Cultures

**DOI:** 10.3390/biomedicines13051234

**Published:** 2025-05-19

**Authors:** Ana B. Perona-Moratalla, Blanca Carrión, Karina Villar Gómez de las Heras, Lourdes Arias-Salazar, Blanca Yélamos-Sanz, Tomás Segura, Gemma Serrano-Heras

**Affiliations:** 1Department of Neurology, General University Hospital of Albacete, Hermanos Falcó, 37, 02008 Albacete, Spain; abperona@gmail.com; 2Research Unit, General University Hospital of Albacete, Laurel, s/n, 02008 Albacete, Spain; blanca_carrion@hotmail.com (B.C.); lourdesas.1230@gmail.com (L.A.-S.); blanca.yelamos.sanz@gmail.com (B.Y.-S.); 3Department of Medicine, Faculty of Medicine, Health and Sports, Universidad Europea de Madrid, 28670 Villaviciosa de Odón, Spain; 4Emergency and Medical Transport Management, Castilla-La Mancha Health Service (SESCAM), 45071 Toledo, Spain; kvillar@sescam.jccm.es; 5Neuroscience Section, Institute of Health Research of Castilla-La Mancha (IDISCAM), 45071 Toledo, Spain; 6Biomedicine Institute of UCLM (IB-UCLM), Faculty of Medicine, University of Castilla-La Mancha, 02008 Albacete, Spain

**Keywords:** Von Hippel-Lindau disease, hemangioblastoma, neurosurgery, cellular proliferative patterns, hypoxia-inducible factors, HIF-1α, HIF-2α, inhibitor, acriflavine, targeted therapy, therapeutic opportunities

## Abstract

**Background:** Von Hippel-Lindau (VHL) disease, a hereditary cancer syndrome, is characterized by mutations in the VHL gene, which result in the stabilization of hypoxia-inducible factors (HIF)-1α and -2α, ultimately leading to the development of highly vascularized tumors, such as hemangioblastomas of the central nervous system (CNS-HBs). The standard treatment for these brain tumors is neurosurgical resection. However, multiple surgeries are often necessary due to tumor recurrence, which increases the risk of neurological sequelae. Thus, elucidation of the proliferative behavior of hemangioblastomas (with the aim of identifying biomarkers associated with tumor progression) and the development of pharmacological therapies could reduce the need for repeated surgical interventions and provide alternative treatment options for unresectable CNS-HBs. Belzutifan (Welireg™), a selective HIF-2α inhibitor and the only FDA-approved non-surgical option, has shown limited efficacy in CNS-HBs, highlighting the need for alternative therapeutic strategies. **Results:** In this study, primary cell cultures were successfully established from CNS-HB tissue samples of VHL patients, achieving a 75% success rate. These cultures were predominantly composed of stromal cells and pericytes. The proliferative patterns of patient-derived HB cell cultures significantly correlated with tumor burden and recurrence in VHL patients. Furthermore, flow cytometry, reverse transcription-PCR, and Western blot analyses revealed marked overexpression of both HIF-1α and HIF-2α isoforms in primary HB cells. In addition, evaluation of the therapeutic potential of acriflavine, a dual HIF-1α/HIF-2α inhibitor, demonstrated reduced HB cells viability, induced G2/M cell cycle arrest, and predominantly triggered necrotic cell death in patient-derived HB cultures. **Conclusions:** These results suggest that the in vitro proliferative dynamics of HB cell cultures may reflect clinical characteristics associated with CNS-HB progression, potentially serving as indicators to predict tumor development in patients with VHL. Furthermore, our findings support the simultaneous targeting of both HIF-1α and HIF-2α isoforms as a promising non-invasive therapeutic strategy.

## 1. Introduction

Von Hippel-Lindau disease (VHL) (OMIM 193300) is a dominantly inherited familial syndrome with an incidence that has been internationally reported to be 1 in 36,000 [1]. This rare disease results from a heterozygous mutation or deletion in the VHL tumor suppressor gene, located on chromosome 3p25-p26 region [2,3]. This genetic alteration predisposes individuals to a variety of neoplasms with complete penetrance and variable expressivity, including retinal and central nervous system hemangioblastomas (CNS-HBs), renal cell carcinoma (RCC), pheochromocytomas (Pheo), paragangliomas, cystic and solid pancreatic lesions (comprising pancreatic neuroendocrine tumors, pNETs), endolymphatic sac tumors, and broad ligament cystadenomas in women or epididymal cystadenomas in men [4,5,6]. Four general VHL phenotypes (type 1, type 2A, type 2B, and type 2C) have been proposed based on the likelihood of developing pheochromocytoma. VHL type 1, which is characterized by a low risk of pheochromocytoma, is typically associated with pathogenic truncating variants that severely disrupt the folding of the VHL protein, as well as with large deletions. In contrast, VHL type 2 is characterized by a high risk of pheochromocytoma, and individuals with this phenotype commonly carry pathogenic missense variants [7,8].

Multiple CNS hemangioblastomas, occurring either synchronously or metachronously, are common in VHL disease, with 60% to 80% of patients developing these tumors, except for those with type 2C VHL. Roughly 80% of these tumors develop in the brain, primarily in the cerebellum and brainstem, while the remaining 20% occur in the spinal cord. Hemangioblastomas typically present as benign, highly vascular tumors, often associated with cyst formation. Neuroimaging studies in patients have revealed that CNS hemangioblastomas (CNS-HBs) exhibit distinct growth patterns, saltatory (72%), linear (6%), or exponential (22%) [9], with increased growth being associated with male sex, symptomatic tumors, hemangioblastoma-associated cysts, and germline VHL intragenic deletions [10]. Despite their clinical relevance, the majority of research on hemangioblastomas has focused on histological and descriptive analyses, which have demonstrated that these tumors are composed of stromal cells and a dense network of blood vessels formed by pericytes and endothelial cells [11,12,13,14,15,16]. However, there is a notable scarcity of studies utilizing primary hemangioblastoma cell cultures. The establishment and characterization of such models could provide valuable insights into tumor proliferation dynamics, including growth rates and the identification of key factors involved in tumor development. This understanding is crucial for unraveling the biological behavior and progression of hemangioblastomas and may aid in identifying biomarkers of aggressiveness or recurrence. Furthermore, primary cultures provide a translational platform for testing antiproliferative agents in a patient-specific context, potentially guiding personalized therapeutic strategies and improving clinical outcomes.

The etiopathogenesis of VHL-associated hemangioblastomas is driven by mutations in the VHL gene, which accounts for the unique microenvironment of these CNS tumors, characterized by a strong influence of hypoxia signaling pathways [17,18,19], as these mutations result in the dysregulation of hypoxia-inducible factors (HIFs) [20,21,22,23,24,25]. HIF-1α and HIF-2α are transcription factors that play key roles in the cellular response to hypoxia [23,26,27,28,29]. Under low oxygen conditions, HIF-1α and HIF-2α are stabilized due to the loss of degradation signals. These factors then dimerize with HIF-β subunits (also known as ARNT) to form functional transcriptional complexes. The HIF-α/β heterodimers translocate to the nucleus, where they bind to hypoxia-responsive elements (HREs) in the promoters of target genes. In VHL-deficient cells, the absence of functional VHL protein prevents the degradation of both HIF-α isoforms [20,25,30,31,32]. As a result, stabilized HIF-1α and HIF-2α lead to the overexpression of various target genes, even under normoxic conditions. Specifically, HIF-1α, which is broadly expressed across most cell types, promotes the upregulation of genes involved in cell survival, glycolysis and angiogenesis. HIF-2α has a more restricted expression pattern, primarily in tissues such as the kidneys, liver, and vasculature, it induces the expression of genes associated with cell proliferation, differentiation, and erythropoiesis [33,34]. In VHL-deficient renal cell carcinoma (RCC), HIF-2α has been demonstrated to be the key oncogenic driver, while HIF-1α also contributes by supporting tumor metabolism [35,36,37]. In the context of VHL-deficient hemangioblastomas, the stabilization of both HIF-1α and HIF-2α has been suggested to drive the tumor’s highly vascularized environment, thereby promoting its growth and survival [19,38]. These findings underscore the critical roles of HIF-1α and HIF-2α in the development, progression, and vascularization of VHL-associated hemangioblastomas.

Although surgical resection remains the standard of care for CNS hemangioblastomas, excision may not be feasible depending on the tumor’s location. In cases where a symptomatic tumor is nonresectable, stereotactic radiosurgery can offer short-term control [39]. The recurrent manifestation of VHL-related tumors, along with the complications associated with repeated surgical interventions or the presence of unresectable tumors, makes CNS-HBs a significant cause of morbidity and mortality in patients with VHL disease. For this reason, in the last decade, the development of pharmacological treatments as a complementary strategy has gained increasing interest.

Recently, Belzutifan (Welireg™), a selective HIF-2α inhibitor, was approved for the treatment of advanced RCC, CNS hemangioblastomas, and pNET associated with VHL [40,41]. The response rate for CNS-HBs, which included both solid and cystic components, was 30%, with only 6% of patients achieving complete responses [42]. This rate is lower than that observed in RCC (49%) and pNET (91%), possibly reflecting the distinct molecular characteristics of these tumors. The involvement of both HIF-1α and HIF-2α in tumor growth and vascularization may explain why selective HIF-2α inhibitors like Belzutifan show limited efficacy in VHL-deficient hemangioblastomas. Hence, we hypothesize that dual inhibition of HIF-1α and HIF-2α may represent a more effective strategy to delay tumor progression and enhance therapeutic outcomes in CNS hemangioblastomas. Consistent with this hypothesis, targeting both HIF-α isoforms with acriflavine, a dual inhibitor that blocks the formation of active transcriptional complexes and reduces HIF transcriptional activity [43,44,45], has previously been shown to promote cytotoxicity in glioblastoma cells, another type of brain tumor [46,47,48], where overexpression of these factors leads to increased proliferation and invasion [49]. These studies demonstrated that the drug is effective against tumor cells in vitro while sparing normal cells. This preferential cytotoxicity, which was statistically significant (*p* ≤ 0.05), underscores the potential of acriflavine to target glioblastoma cells while preserving healthy cell cultures. Furthermore, in vivo animal experiments have shown that acriflavine exhibits a favorable safety profile when administered intraocularly or intraperitoneally [45]. However, systemic administration has been associated with some toxicity, usually measured as weight loss [50]. To address this problem, several research groups have shown that encapsulating acriflavine in various delivery systems, such as wafers [46], implants [48], or lipid nanocapsules [50], is a safe strategy for use in oncology, whether for systemic or local treatment, including intrathecal administration. In the latter case, acriflavine has been shown to preserve the viability of both neuronal and glial cells [46].

This study aims to analyze the growth profile, as well as HIF-1α and HIF-2α mRNA and protein levels, in an in vitro CNS hemangioblastoma model based on patient-derived primary cell cultures. These cultures, which better preserve the genetic and phenotypic characteristics of the original tumor, enabled the assessment of proliferation rates and their correlation with key clinical variables, including the presence of multiple or recurrent tumors. Given that primary HB cells also provide a more reliable platform for testing potential therapeutic interventions, the effects of simultaneous HIF-1α and HIF-2α inactivation on cell survival, cell cycle progression, and induction of cell death were further evaluated using acriflavine, a dual HIF-α inhibitor. This study seeks to enhance our understanding of the growth dynamics of VHL-related hemangioblastomas, with a particular focus on determining the effect of dual HIF-α isoform inhibition on tumor progression, which could potentially contribute to the development of novel pharmacological strategies.

## 2. Materials and Methods

### 2.1. Patient Recruitment, Clinical Data Collection, and Tissue Sampling of CNS Hemangioblastomas and Gliomas

Patients diagnosed with VHL disease who underwent neurosurgical procedures at the Jiménez Díaz Foundation Hospital, Madrid (Spain) for CNS-HBs were offered participation in the study. A total of 12 excess resected hemangioblastoma tissue samples were collected from 11 patients (one patient underwent two surgeries). Concurrently, glioma tumor samples, for use as controls in cellular and molecular experiments, were obtained post-surgery from 3 patients diagnosed with astrocytoma, gliosarcoma, and glioblastoma, respectively, by the Neurology and Neurosurgery departments of the General University Hospital of Albacete, Spain. Written informed consent was obtained from all participants in accordance with the 1964 Declaration of Helsinki and its later amendments. All procedures were approved by the Human Ethics Committees of both hospitals (Ethical approval numbers: 2014/2 and 03/2014). Sociodemographic data (sex and age) and clinical information, including genetic profile (deletion, nonsense, or missense mutation of the VHL gene), central nervous system location of the hemangioblastoma, and synchronous or metachronous appearance of other lesions assessed by radiographic examinations, were collected (see Table 1). Immediately after resection, fresh tissue samples were placed in Earle’s Balanced Salt Solution (EBSS, #24010043, Thermo Fisher Scientific S.L.U., Madrid, Spain) cell culture medium, which maintains cell viability, and transported at room temperature to the Research Unit, General University Hospital of Albacete, within 6 h.

### 2.2. Protocol for Establishing Primary Cell Cultures of CNS Hemangioblastomas and Gliomas and Use of Immortalized Cell Lines

Upon receipt of tissue samples from 11 CNS hemangioblastomas and 3 gliomas (astrocytoma, gliosarcoma, and glioblastoma) at the Hospital of Albacete, we applied the method for the establishment and growth of patient-derived primary cell cultures, as previously described by our group [51]. It is worth noting that these primary cultures were obtained for the first time, as no studies with HB cells derived from patients had been earlier described in the literature. Briefly, tumor samples were washed multiple times with Phosphate-Buffered Saline (PBS), pH:7.4 (#10010023, Thermo Fisher Scientific) and cut into 1 mm^3^ pieces. Tissue pieces were then transferred using sterile tweezers to clean 60-mm cell culture dishes (#83.3901.500, Sarstedt S.A.U., Barcelona, Spain) and subjected to enzymatic digestion with an equal volume of collagenase type I (#117018029, Thermo Fisher Scientific) and dispase type II (#17105041, Thermo Fisher Scientific), both at a final concentration of 1 mg/mL in EBSS, for 45 min at 37 °C. Subsequently, the pieces were completely disaggregated by gentle pipetting and further digested with trypsin for 15 min at 37 °C. The minced samples were then centrifuged, and the resulting cell pellets were resuspended in growth medium (RPMI 1640, #A4192101, Thermo Fisher Scientific) supplemented with 20% fetal bovine serum (FBS-heat inactivated/gamma irradiated, #A5669201, Thermo Fisher Scientific), 100 U/mL penicillin/streptomycin (P/S, 10.000 U/mL, #15140122) and 5 mM L-glutamine (#A2916801, Thermo Fisher Scientific), and incubated at 37 °C in a humidified 5% CO_2_ incubator. The medium was replaced every 72 h until the cultures reached confluence. Cells were then detached using 0.25% trypsin and subcultured in cell plates with fresh proliferative medium for subsequent cellular and molecular experiments.

In addition to the primary glioma cultures, immortalized cell lines were also used as controls. SW48 (derived from a human colorectal adenocarcinoma) and ARPE-19 (derived from the human retinal pigment epithelium) cell lines were obtained from the American Type Culture Collection, Manassas, VA, USA (ATCC: #CCL-231 and #CRL-2302, respectively). SW48 and ARPE cells were maintained in the same media conditions as those for patient-derived cell cultures: RPMI 1640 medium supplemented with 20% FBS, 1% P/S, and 5 mM L-glutamine at 37 °C in a humidified atmosphere containing 5% CO_2_.

### 2.3. Cell Profiling and Analysis of HIF-α Isoform Protein Expression: Flow Cytometry and Western Blot

First, HIF-1α and HIF-2α protein expression was determined by flow cytometry in the 9 successfully established primary hemangioblastoma cultures (HB2, HB3, HB4, HB7, HB8, HB9, HB10, HB11, and HB12), derived from VHL patients who underwent neurosurgical procedures and whose clinical and tumor characteristics are shown in Table 1 and Table 2. Additionally, 5 control samples were included: three primary glioma cell cultures (astrocytoma, gliosarcoma, and glioblastoma) and 2 established cell lines (SW48 and ARPE-19). All cells were harvested at a density of 0.3 × 10^6^ cells and centrifuged. The resulting cell pellets were resuspended in PBS and incubated with fluorophore-conjugated antibodies that specifically recognize the membrane markers of each cell type present in HB (phenotypic characterization): anti-CD99-APC (stromal cells) (#99A-100T-cloneHI156, Immunostep, Salamanca, Spain), anti-NG2-FITC (pericytes) (#18646693-clone:7.1, Thermo Fisher Scientific), and anti-CD34-PE-Cy7 (endothelial cells) (#34PC7-100T-CUSTOM-clone 581, Immunostep) for 2 h at room temperature in the dark. Following a wash, the cells were fixed with Intracellular Reagent A (#INTRA-100T, Immunostep) for 15 min, washed again with PBS, and permeabilized with Intracellular Reagent B (#INTRA-100T, Immunostep). Subsequently, the cells were incubated with either anti-HIF-1α-PE (#HIF1PE-015mg-CUSTOM-cloneH1alpha67, Immunostep) or anti-HIF-2α-PE (#HIF2PE-01mg-CUSTOM-clone ep190b, Immunostep) antibodies for 1 h in the dark. Finally, the cells were centrifuged, the supernatant was removed, and the cells were resuspended in 400 µL of PBS for analysis using a FACS Canto II flow cytometer with FACS DIVA software v6.1.3. These antibodies yielded data on the percentage of cells expressing HIF-1α or HIF-2α in the primary cultures of CNS-HB and gliomas, and in the immortalized human cell lines. Moreover, in the case of the HB cells, it was possible to determine the cellular distribution of endothelial, pericyte, and stromal cells in the primary cultures and the HIF-α isoform expression profile among the different cellular components.

In parallel, total proteins were extracted using RIPA buffer (#R0278-500 mL, Merck España, Barcelona, Spain) from 9 primary hemangioblastoma cell cultures (HB2, HB3, HB4, HB7, HB8, HB9, HB10, HB11, and HB12), 2 primary glioma cell cultures (astrocytoma-AST and glioblastoma-GB), and the SW48 colon cancer cell line, all grown to a density of 1.2–1.5 × 10^6^ cells (equivalent to 80% confluence in 100-mm cell culture dishes, # 83.3902.300, Sarstedt). The protein concentration was determined using a BCA assay. Subsequently, protein extracts (20–30 µg) from 7 HB cultures (HB2, HB3, HB4, HB7, HB8, HB9, and HB10) were separated by denaturing SDS-PAGE and transferred to PVDF membranes (Hybond-C Extra, Amersham Biosciences Europe GmbH, Barcelona, Spain). HB11 and HB12 were excluded from the Western blot analysis due to insufficient protein yield. A Spectra™ Multicolor Broad Range Protein Ladder (#26634, Thermo Fisher Scientific) was utilized as a molecular weight marker. The primary antibodies used were anti-HIF-1α (clone 28b, #sc-13515HRP, Santa Cruz Biotechnology, Inc., Heidelberg, Germany) and anti-HIF-2α (clone 190b, #sc-13596HRP, Santa Cruz), and anti-βtubulin (clone D-10, # sc-5274) as a loading control. Chemiluminescence detection was achieved using SuperSignal™ West Dura (#34075, Thermo Fisher Scientific) on a Luminescent Image Analyzer LAS-mini 4000 system (Fujifilm, Tokyo, Japan).

### 2.4. RNA Isolation and Real-Time RT-PCR Analysis

The mRNA levels of HIF-1α and HIF-2α were determined as follows: Total RNA was extracted using the QIAshredder and RNeasy Mini Kit (#79654 and #74104, QIAGEN Iberia, S.L., Barcelona, Spain) from 9 primary hemangioblastoma cell cultures (HB2, HB3, HB4, HB7, HB8, HB9, HB10, HB11, and HB12), 2 primary glioma cell cultures (astrocytoma and glioblastoma), and the SW48 colon cancer cell line, all grown to a density of 1.2–1.5 × 10^6^ cells (equivalent to 80% confluence in 100-mm cell culture dishes, #83.3902.300, Sarstedt). RNA concentration was measured by spectrophotometry using a NanoDrop ND-100. Subsequently, cDNA was synthesized from 1 µg of total RNA extracted from 7 HB cultures (HB2, HB3, HB4, HB7, HB8, HB9, and HB10) using the RevertAid™ Minus First Strand cDNA Synthesis Kit (#K1632, Thermo Fisher Scientific), which includes dNTPs, random hexamers, oligo(dT) primers, and reverse transcriptase. HB11 and HB12 were excluded from the analysis due to insufficient RNA yield. Real-time PCR was then performed on the cDNA using the SYBR Green PCR Kit (#4385612, Life Technologies S.A., Madrid, Spain) and a STEP ONE PLUS system (Applied Biosystems). mRNA levels were normalized to tubulin and quantified using the 2-ΔΔCT method to determine relative gene expression. The sequences of the primers used were:

HIF-1α: Forward:5′-ACTTCTGGATGCTGGTGATT-3′, Reverse:5′-TCCTCGGCTAGTTAGGGTAC-3′

HIF-2α: Forward:5′-CAACCTCAAGTCAGCCACCT-3′, Reverse:5′-TGCTGGATTGGTTCACACAT-3′

β-Tubulin: Forward:5′-CTTCGGCCAGATCTTCAGAC-3′, Reverse: 5′-AGAGAGTGGGTCAGCTGGAA-3′

### 2.5. Cell Viability

Cell viability and proliferative activity were assessed using a colorimetric assay based on MTT (3-(4,5-dimethylthiazol-2-yl)-2,5-diphenyltetrazolium bromide) reagent (#M2128, Merck). 7 primary hemangioblastoma cell cultures, exhibiting low to moderate (HB2, HB3, HB4, HB8) and high (HB7, HB9, HB10) HIF-1α/2α overexpression, were seeded at a density of 0.4 × 10⁵ cells per well in 24-well culture plates (#83.3922.005, Sarstedt). Cells were incubated in 500 µL of RPMI 1640 medium supplemented with 20% FBS, 100 U/mL penicillin-streptomycin (P/S), and 5 mM L-glutamine, and subsequently treated with increasing concentrations (1 µM, 2.5 µM, 5 µM, 10 µM, 25 µM, 50 µM, 100 µM) of purified acriflavine (≥95% HPLC; a mixture of trypaflavines: 3,6-diamino-10-methylacridinium chloride and 3,6-diamino-3-methylacridine hydrochloride, and proflavine: 3,6-diaminoacridine hydrochloride) (#SML3350, Merck). Following 24-, 48-, and 72-h incubations with acriflavine, MTT solution (5 mg/mL) was added and the cells were incubated for 1 h at 37 °C. The medium was subsequently aspirated, and a solubilization solution was added to dissolve the resulting formazan crystals. The MTT assay relies on the reduction of MTT to formazan by metabolically active cells. The amount of formazan produced, as measured by the absorbance at 570 nm using a microplate reader (SPECTROstar Omega, Software Version 5.10, BMG LABTECH), is directly proportional to the number of viable cells. The percentage of cell viability was then calculated relative to untreated control cells

### 2.6. Cell Cycle Analysis and Apoptosis/Necrosis Assays

7 primary hemangioblastoma cell cultures, exhibiting low to moderate (HB2, HB3, HB4, HB8) and high (HB7, HB9, HB10) HIF-1α/2α overexpression were seeded in 60-mm cell culture dishes and grown to a density of 0.3 × 10^6^ cells, then exposed to 2.5 µM and 25 µM acriflavine or PBS-vehicle for 24 h. Subsequently, cells were harvested, washed with PBS, fixed, and permeabilized using a cold fixative solution containing ethanol to preserve their DNA content and allow access of staining reagents to the DNA. The cells were then incubated with propidium iodide (PI) (10 mg/mL) (#P1304MP, Thermo Fisher Scientific), a DNA-binding dye that intercalates with DNA in a stoichiometric manner, meaning the amount of dye bound is directly proportional to the amount of DNA. To determine the distribution of cells across the cell cycle phases (G0/G1, S, and G2/M), the PI-stained cells were analyzed using flow cytometry. The collected data were analyzed using FACS DIVA software to determine the percentage of cells in each phase of the cell cycle and to generate histograms.

In addition, to assess acriflavine-triggered cell death, these HB cell cultures (at a density of 0.3 × 10^6^ cells per sample) were treated in duplicate with acriflavine at doses of 2.5 µM and 25 µM, or with a PBS-vehicle control, for 48 h. Following treatment, cells were harvested, centrifuged, and resuspended in 450 µL of 1X Annexin V binding buffer (#BB10X-50 mL, Immunostep). The cells were then stained with 10 µL of Annexin V conjugated to Dy-634 (#ANXDY-200T, Immunostep) and 40 µL of PI (10 mg/mL) for 1 h in the dark. Cell death analysis was performed using a FACS Canto II flow cytometer. Early apoptotic cells (Annexin V-positive, PI-negative), late apoptotic cells (Annexin V-positive, PI-positive), and necrotic cells (Annexin V-negative, PI-positive) were quantified and represented using FACS DIVA software.

### 2.7. Statistical Analysis

Data were analyzed using IBM SPSS Statistics 22. Statistical significance was assessed using Student’s *t*-test for two-group comparisons to evaluate the transcriptional activity of HIF-1α or HIF-2α in each primary HB cell type relative to SW48 carcinoma cells. Additionally, Chi-square tests were employed to explore potential associations, while Cramer’s V was used to quantify the strength of the relationships between in vitro growth patterns (classified by doubling time and lifespan as either slow-growing/short-term or moderate-growing/mid-term), the expression levels of hypoxia-inducible factors HIF-1α and HIF-2α (measured through flow cytometry, transcript, and protein analyses, and categorized as low-to-moderate or high overexpression), and key clinical indicators, including the extent of tumor involvement (single versus multiple lesions), recurrence within a four-year period, and overall disease severity. Statistical analysis was also performed on the cell viability assay (MTT) using Student’s *t*-test for two-group comparisons to evaluate each concentration of acriflavine treatment compared to untreated cells at each incubation time point (24, 48, and 72 h).

## 3. Results

### 3.1. Descriptive Study of Clinical Features and VHL Mutations in a Cohort of Patients with CNS Hemangioblastomas

The study cohort consisted of 11 VHL patients who underwent CNS hemangioblastoma resection. Our study is based on a total of 12 HB samples, as one patient experienced two surgeries, providing tumor specimens from both procedures (HB3 and HB7). Disease-related characteristics of the enrolled patients with VHL are shown in Table 1 and Table 2.

The cohort consisted of 5 males and 6 females, with ages ranging from 13 to 57 years. Genetic analysis revealed that mutations in the VHL gene were identified in all of the patients, with missense mutations being the most frequent (7/11, 63.6%). Other mutations included truncating/frameshift mutations (2/11, 18.2%) and deletions (2/11, 18.2%). At the time of surgery, hemangioblastoma locations varied, with most lesions found in the spinal cord region—including radicular/cauda equina and the medulla oblongata (7 out of 12 surgeries, 58.3%), followed by the cerebellum (4/12, 33.3%) and the temporal lobe (1/12, 8.3%) (Table 1). Figure 1 illustrates the surgical removal of HB8 located in the cerebellum. As is commonly observed with this type of tumor, the lesion is highly vascularized. The images show different stages of the surgical procedure, highlighting the identification and initial dissection steps, making the tumor accessible for subsequent complete resection. A total of 7 patients (63.6%) presented with a single CNS-HB lesion, while 5 patients (45.5%) had multiple tumors (>3 lesions). Follow-up after CNS-HB resection revealed recurrence within four years in 6 cases (HB3, HB4, HB5, HB9, HB10, and HB12), representing 54.5% of the cohort (Table 2). Additionally, clinical record reviews indicated that patients developed a variety of other VHL-associated tumors, either concurrently or at different time points (Table 1). Retinal tumors were very common, observed in 8 of the 11 patients (72.7%). Renal cell carcinoma (RCC) was observed in 5 patients (45.5%). Other tumors included pancreatic cysts (3 patients), pancreatic neuroendocrine tumors (PNET) (2 patients), pheochromocytomas (Pheo) (2 patients), and endolymphatic sac tumors (ELST) (1 patient). Based on available clinical data at the time of writing this article and the genetic characteristics of the patient cohort, we could potentially classify the VHL patients into different subtypes. Type 1 von Hippel–Lindau (VHL) disease is typically associated with truncating mutations and a high risk of central nervous system hemangioblastomas (CNS-HBs) and renal cell carcinoma (RCC), but a low risk of pheochromocytomas (Pheo). In contrast, Type 2 VHL results from missense mutations and includes three subtypes: Type 2A (high risk of CNS-HB and Pheo, low risk of RCC), Type 2B (high risk of CNS-HB, RCC, and Pheo), and Type 2C (risk restricted to Pheo). Thus, HB1, HB2, HB6, and HB8 are consistent with Type 1 VHL, as they harbor truncating or deletion mutations and are associated with CNS-HBs and RCC, but no Pheo. HB10 best fits Type 2B, given the presence of a missense mutation (Thr133Pro) along with CNS-HB, RCC, Pheo, and pancreatic cysts, representing a complete VHL phenotype. HB9 may also correspond to Type 2B, as it carries a missense mutation (Asn78Ser) and presents with CNS-HB, Pheo, pancreatic cysts, and pancreatic neuroendocrine tumor (pNET), though RCC is not reported. HB4 and HB5, both with the Arg167Trp missense mutation, are consistent with Type 2A, exhibiting CNS and retinal HBs as well as pNET, with no RCC or Pheo reported. Finally, HB3, HB7, HB11, and HB12 also carry missense mutations and show multiple CNS or retinal HBs; however, none present with Pheo, and only HB12 has pancreatic cysts. These cases may represent Type 2A, though their classification remains less definitive due to the absence of Pheo or RCC in most samples. Regarding disease aggressiveness, VHL exhibits significant clinical heterogeneity, and currently, there are no standardized criteria for grading disease severity across patients. This variability is influenced by factors such as incomplete penetrance, diverse mutation types, the number and recurrence of retinal and CNS hemangioblastomas (often requiring repeated surgical interventions with outcomes dependent on tumor location and neurosurgeon experience), and the presence of other tumors. Consequently, patients presenting with multiple CNS-HBs, recurrent tumors, and additional manifestations such as RCC and pheochromocytoma may be considered as having a more severe disease phenotype and an unfavorable prognosis. In our cohort, 4 patients (33.3%) were classified as having a severe/unfavorable course, 4 as moderate/average, and 3 (25%) as mild/favorable (Table 2). These data provide a descriptive overview of the variability in the behavior of VHL-associated hemangioblastoma cases, the presence of other tumors, and clinical outcomes among patients.
biomedicines-13-01234-t001_Table 1Table 1Clinical characteristics of the enrolled patients with VHL disease.HBSamplesSexAgeMutation in Vhl GenCNS LocationOther Tumors (VHL-Related)Establishment of Primary Culture***HB1***Female39Truncating/frameshift mutation in exon 1Spinal cord (Radicular/Cauda)Pancreatic serous cystoadenoma, RCC-***HB2***Male13DeletionMedulla oblongataRetinal HB, RCC√***HB3 ****Male55Missense mutationTemporal lobeRetinal HB√***HB4 *****Female49Missense mutation (arg167-trp) in exon 3Spinal cord (Radicular/Cauda)Cerebellar HB, retinal HB, pNET√***HB5 *****Male19Missense mutation (arg167-trp) in exon 3Spinal cord (Radicular/CaudaCerebellar HB, retinal HB-***HB6***Female37Truncating/frameshift mutationSpinal cord (C6-C7)RCC-***HB7 ****Male56Missense mutationSpinal cord (Radicular/Cauda)Retinal HB√***HB8***Male46DeletionCerebellumRCC√***HB9***Female49Missense mutation (asn78-ser) in exon 1Medulla oblongata/MedularCerebellar HB, retinal HB, Pancreatic cysts, pNET, Pheo√***HB10***Male49Missense mutation (thr133-pro) in exon 3CerebellumRetinal HB, RCC, Pheo, pancreatic cysts√***HB11 ******Female57Missense mutation(leu184-pro) in exon 3CerebellumRetinal HB√***HB12 ******Female29Missense mutation (leu184-pro) in exon 3CerebellumRetinal HB,Spinal HB,pancreatic cysts√* CNS hemangioblastomas collected from the same patient, ** mother and son *** aunt and niece. Abbreviations: RCC: Renal Cell Carcinoma, HB: Hemangioblastoma, pNET: Pancreatic Neuroendocrine Tumor, Pheo: Pheochromocytoma. √: YES, -: NO.

### 3.2. Patient-Derived HB Cell Cultures Show High Expression Levels of Both HIF-1α and HIF-2α Isoforms

To better understand the biology and growth dynamics of VHL-associated CNS hemangioblastomas (CNS-HBs), we analyzed the in vitro proliferation rate and assessed HIF-α expression levels in patient-derived cell cultures using cellular and molecular techniques. To this end, we first proceeded with the establishment of primary cultures from CNS-HB samples collected post-surgery, as described by our group [51] and briefly outlined in the Section 2. Notably, 9 primary cell cultures derived from a total of 12 CNS hemangioblastoma tumor samples, obtained immediately post-surgical resection and transported to the cell culture laboratory in EBSS solution within 6 h, were successfully established (75% success rate), allowing subsequent in vitro investigations. These patient-derived HB cell cultures grew as a monolayer and predominantly displayed a fusiform morphology, with a smaller proportion presenting a polygonal shape (Figure 2a). They reached 80% confluence, equivalent to approximately 1.2–1.5 × 10^−6^ cells, within 3 to 4 weeks. As shown in Table 2, the primary cells exhibited doubling times ranging from 86 to 96 h and were capable of proliferating until the 11th to 14th generation, after which cell growth ceased and cultures died within 6 to 8 weeks. Based on doubling time and lifespan analysis, cellular proliferation patterns were classified as slow-growing (>90 h) with short-term viability (<12 generations) in 4 cases, and moderate-growing (<90 h) with mid-term viability (>12 generations) in 5 cases. While the cell morphology was similar, the growth rate was slower than that reported in previous work [50]. In contrast, the lifespan of our primary HB cells, even those that exhibited limited proliferative capacity, was extended, allowing proliferation for more than two weeks. This exceeded the duration observed by those authors, potentially due to differences in culture media and materials. On the other hand, despite all these tumor samples being carefully maintained under the same appropriate conditions to ensure their viability for primary cells establishment, HB1, HB5, and HB6 were unable to grow and quickly entered the senescence phase, dying a few days later (Figure 2b). Although these CNS hemangioblastomas were all derived from a spinal cord location, it is unlikely that the failure of hemangioblastoma growth was due to this, as 2 other samples, HB4 and HB7, were removed from the spinal cord and successfully grew to confluence. The challenge in establishing these three cultures was probably related to the small initial size of the samples, which measured less than 0.3 cm × 0.4 cm, while the successfully samples were larger than 0.5 cm × 0.5 cm.

Subsequently, the 9 continuously growing HB cell cultures were incubated with specific membrane markers for stromal cells, pericytes, and endothelial cells and analyzed by flow cytometry to confirm the presence of well-known cell components of CNS hemangioblastomas. As expected, the results showed that the primary HB cultures contained roughly 35–45% stromal cells, 25–40% pericytes, and 5–15% endothelial cells.

This cellular analysis technique was further used to determine the percentage of cells expressing HIF-1α and HIF-2α in the primary HB cultures (Figure 3a,b), compared to those present in primary cultures derived from other brain tumors (astrocytoma, gliosarcoma, and glioblastoma), as well as in the SW48 (colorectal adenocarcinoma cell line) and ARPE-19 (immortalized human retinal pigment epithelium) cell lines. As shown in Figure 3, the percentage of cells positive for HIF-1α and HIF-2α varied among the different primary hemangioblastoma cells, indicating some heterogeneity in hypoxia-related protein expression. Nevertheless, in all patient-derived HB cell cultures, the levels of HIF-1α and HIF-2α expression were significantly higher than in non-hemangioblastoma cells, with a slightly higher proportion of cells showing positivity for HIF-1α compared to HIF-2α. Specifically, HB7 and HB9, derived from the spinal cord and medulla oblongata, respectively, and HB10, HB11, and HB12, derived from the cerebellum, exhibited the highest percentages of cells with intracellular synthesis of HIF-1α (85–90%) and HIF-2α (80–85%). In contrast, HB3, cultured from the temporal lobe, showed the lowest levels of both markers (22.1% for HIF-1α and 14.7% for HIF-2α). The remaining cultures, HB2, HB4, and HB8, derived from the medulla oblongata, spinal cord, and cerebellum, respectively, presented intermediate expression levels, displaying 40–50% and 35–45% of HIF-1α and HIF-2α positive cells. When comparing these values with non-hemangioblastoma cell lines, the expression of HIF-1α and HIF-2α was markedly lower in gliosarcoma cells (3% and 1.7%, respectively) and SW48 colorectal carcinoma cells (8.1–1.9% for HIF-1α and 9.1–4.4% for HIF-2α), while increased levels were observed in astrocytoma, glioblastoma, and ARPE-19 cell lines. Although the crucial role of hypoxia-inducible factors in mediating tumor progression in astrocytic tumors or in adapting to the high metabolic demand of the retina has been reported, these cells did not reach the high levels of HIF positivity detected in most HB cells in this study (Figure 3a). Subsequently, we found that the HIF-1α expression profile among the different cellular components of all primary HB cultures was similar, with stromal cells showing the highest positivity for the protein, reaching 35–45%, followed by pericytes (20–30%). Endothelial cells exhibited lower expression across all cultures, ranging from 2.2% to 15%. Likewise, HIF-2α was mainly expressed in stromal cells and pericytes, with low levels detected in endothelial cells (Figure 3b). These results indicate an elevated proportion of HIF-expressing cells across all patient-derived HB cultures, with a particularly high prevalence in stromal and pericytic cell populations, suggesting a potential role for these cell populations in the hypoxic response of hemangioblastomas and thereby in tumor progression.

Next, to confirm the high percentage of HIF-positive cells observed in HB cells by flow cytometry, we performed real-time RT-PCR and Western blot analysis. Specifically, mRNA and protein levels of HIF-1α and HIF-2α were assessed in seven primary HB cell cultures (HB2, HB3, HB4, HB7, HB8, HB9, and HB10), as HB11 and HB12 did not yield sufficient RNA or protein extracts for molecular analyses and subsequent acriflavine treatment. As depicted in Figure 4a, HB7, HB9, and HB10 demonstrated the highest transcriptional levels of HIF-1α and HIF-2α, approximating a 5-fold increase compared to the levels observed in SW48 colon cancer cells, which served as a control. HB2, HB3, and HB8 also showed significantly elevated expression, with values surpassing by 3-fold those of the SW48 cell line. By contrast, HB4 displayed lower levels, remaining proximate to baseline. Primary cultures of astrocytoma and glioblastoma exhibited moderate expression of HIF- α isoforms, though not at the same magnitude as in the majority of HB cultures. Western blot results were consistent with the mRNA expression data, showing markedly elevated levels of HIF-1α and HIF-2α proteins in HB7, HB9, and HB10, whereas HB2, HB3, HB4, and HB8 exhibited low to moderate expression of these hypoxia-inducible factors (Figure 4b). Both experimental approaches revealed a slight to moderate trend towards increased HIF-1α expression.

Together, these cellular and molecular assay results indicate that all patient-derived primary HB cell cultures from individuals with VHL disease exhibit marked overexpression of HIF-1α and HIF-2α. Expression levels ranged from low to moderate in 44.5% of cases and were high in 55.5%, regardless of the tumor’s localization within the CNS, and consistently exceeded those observed in astrocytic tumors, which are well known for upregulating hypoxia-responsive factors and genes. These findings suggest that the robust hypoxic response observed in hemangioblastomas, as evidenced by elevated HIF-α isoform levels, may be associated with microenvironmental adaptation, as well as tumor development and progression.

### 3.3. Proliferative Patterns of Primary HB Cell Cultures Significantly Correlate with Tumor Burden and Recurrence in VHL Patients

With the aim of identifying potential cellular characteristics that may reflect or predict the clinical behavior of CNS hemangioblastomas, and thereby contribute to a better understanding of tumor heterogeneity and progression in VHL disease, we investigated whether specific cellular and molecular phenotypes correlate with clinical features of VHL-associated CNS hemangioblastomas, as summarized in Table 2.

Specifically, chi-square tests were used to assess associations, and Cramer’s V was applied to evaluate the strength of the relationships between in vitro growth patterns (based on doubling time and lifespan, categorized as slow-growing/short-term or moderate-growing/mid-term), the expression levels of hypoxia-inducible factors (HIF-1α and HIF-2α, determined by flow cytometry, transcript, and protein analyses, and classified as low-to-moderate or high overexpression), and key clinical variables, including tumor burden (single vs. multiple lesions), recurrence within four years, and overall disease severity.

The analysis revealed no statistically significant association between cellular proliferative patterns and either HIF-1α/HIF-2α overexpression (χ^2^ = 2.723, *p* = 0.206; Cramer’s V = 0.550) or overall disease severity (χ^2^ = 3.600, *p* = 0.165; Cramer’s V = 0.632). These findings are likely due to the fact that, as previously mentioned, all primary HB cultures consistently exhibited high levels of HIF-1α and HIF-2α—i.e., overexpression—resulting in no significant differences between cultures. This also suggests that additional cellular factors may contribute to the proliferation rate of HB cells. Moreover, since the aggressiveness of VHL disease is influenced by multiple factors unrelated to hemangioblastomas—such as the type of VHL mutation, surgical complications, and the presence of other tumors—it was expected that no direct correlation would be found between this clinical parameter and the cellular behavior of HB cells. Notably, a significant association was observed between cellular proliferation patterns and tumor burden (i.e., the number of CNS-HBs at the time of surgery) (χ^2^ = 5.760, *p* = 0.048). The Cramer’s V value of 0.800 indicated a very strong relationship, with moderate-growing and mid-term viability cultures more frequently associated with multiple tumors. Similarly, cellular proliferation patterns were significantly associated with tumor recurrence within four years (χ^2^ = 5.625, *p* = 0.048), with moderate-growing and mid-term viability cultures more commonly linked to recurrence. The strength of this association was also high (Cramer’s V = 0.791). In contrast, due to the lack of association between proliferation profiles and HIF-1α/HIF-2α overexpression, this molecular variable showed no significant associations with tumor burden (χ^2^ = 1.102, *p* = 0.524), tumor recurrence (χ^2^ = 0.900, *p* = 0.524), or disease severity (χ^2^ = 0.900, *p* = 0.635).

Overall, HB cells exhibiting moderate to high growth capacity and prolonged lifespan were significantly associated with the presence of multiple hemangioblastomas at the time of surgery (>3 lesions) and with recurrence within four years, suggesting that in vitro proliferative behavior may reflect clinical features specifically related to CNS-HB progression, such as tumor burden and the development of recurrent lesions.
biomedicines-13-01234-t002_Table 2Table 2Cellular and molecular features of primary HB cultures, tumor-related parameters, and disease aggressiveness in VHL patients.HBSamplesDoublingTime (Hours)Life Span(Nº of Generations *)CellularProliferativePatterns †HIF-1/2αOverexpressionHBs(Nº at Surgery)CNS-HB Recurrence(<4 Years)Disease Severity/Prognosis***HB1***ndndNo growthnd>3-SEVERE/UNFAVORABLETruncatingNon-recurrent HB, RCC***HB2***9411Slow growth Short-term viabilityLow-moderate1-SEVERE/UNFAVORABLEDeletion,Non-recurrent HB, RCC***HB3***9012Slow growth Short-term viabilityLow-moderate1√MILD/FAVORABLEMissense mut,Non-Recurrent HB***HB4***8813Moderate growth Mid-term viabilityLow-moderate>3√MODERATE/AVERAGEarg167-trp,Recurrent HB, pNET***HB5***ndndNo growthnd1√MODERATE/AVERAGEarg167-trp,Recurrent HB***HB6***ndndNo growthnd1-SEVERE/UNFAVORABLETruncating,Non-recurrent HB, RCC***HB7***8713Moderate growth Mid-term viabilityHigh1√MILD/FAVORABLEMissense mut,Non-recurrent HB***HB8***9611Slow growth Short-term viabilityLow-moderate1-SEVERE/UNFAVORABLE:Deletion,Non-recurrent HB, RCC***HB9***8913Moderate growth Mid-term viabilityHigh>3√MODERATE/AVERAGEasn78-ser,Recurrent HB, Pheo, Pancreatic cysts***HB10***8814Moderate growth Mid-term viabilityHigh>3√SEVERE/UNFAVORABLE:thr133-pro,Recurrent HB, RCC, Pheo, Pancreatic cysts***HB11***9012Slow growth Short-term viabilityHigh1-MILD/FAVORABLEleu184-pro,Non-recurrent HB***HB12***8614Moderate growth Mid-term viabilityHigh>3√MODERATE/AVERAGEleu184-pro, Recurrent HBPancreatic cystAbbreviations: CNS-HB: hemangioblastomas of central nervous system, RCC: Renal Cell Carcinoma, pNET: Pancreatic Neuroendocrine Tumor, Pheo: Pheochromocytoma. nd: not determined (due to failure to establish primary cultures). √: YES, -: NO. * Life span refers to the number of proliferative generations reached by the primary culture in the exponential phase until cell growth ceases. † Cellular proliferative patterns: Slow growth: doubling time > 90 h, Moderate growth: doubling time < 90 h, Short-term viability: <12 generations of life span, Mid-term viability: >12 generations of life span.

### 3.4. Treatment with Acriflavine Results in Decreased Viability, Cell Cycle Arrest, and Increased Necrosis in Patient-Derived HB Cell Cultures

To evaluate novel pharmacological approaches for limiting the proliferation and growth of CNS hemangioblastomas, patient-derived primary tumor cultures were subsequently incubated with acriflavine, a drug that blocks the binding of HIF-α isoforms to the β subunit, thereby inhibiting HIF transcriptional activity. After treating HB cells, exhibiting low-moderate (HB2, HB3, HB4 and HB8) to high (HB7, HB9, and HB10) overexpression of HIF-1α and HIF-2α, with increasing concentrations of acriflavine (1 µM to 100 µM), optical microscopy analysis revealed significant structural and morphological alterations (Figure 5a). Cells became rounded and clustered in response to treatment with 5–10 µM of the drug for 24 h. At higher concentrations (25–100 µM), cells exhibited signs of severe stress, including shrinkage, loss of adhesion, and cell death. Using the MTT assay, which measures metabolic activity as an indicator of cell viability, we found that acriflavine in the culture medium led to a notable, dose- and time-dependent reduction in cell growth in these 6 primary HB cultures. Acriflavine significantly reduced cell survival and proliferation, with viability dropping below 50% at low concentrations (2.5–5 µM) within 48–72 h, highlighting its strong cytotoxic effect (Figure 5b). These results, along with microscopic observations, indicate that acriflavine effectively suppresses hemangioblastoma growth in vitro.

To gain further insight into the effect of acriflavine on cell cycle regulation and cell death, primary HB cells with moderate to high HIF-α overexpression were treated with 2.5 and 25 µM of the inhibitor for 24 h. The cells were then stained with the DNA-intercalating agent propidium iodide and analyzed by flow cytometry. The results showed that blocking HIF transcriptional activity induced cell cycle arrest in the G2/M phase, with a considerable increase in cells containing duplicated DNA that struggled to complete mitosis and continue dividing. Specifically, the percentage of cells in the G0/G1 phase (resting or pre-synthesis DNA phase) decreased from 72.3 ± 0.85% in untreated cells to 52.7 ± 0.71% and 49.9 ± 2.69% in cells treated with 2.5 µM and 25 µM acriflavine, respectively. Meanwhile, the S phase (DNA replication phase) remained relatively stable, with 8.25 ± 1.06%, 7.55 ± 1.34%, and 8.75 ± 1.63% in untreated, 2.5 µM-, and 25 µM-treated cells, respectively. Interestingly, a significant accumulation of cells was observed in the G2/M phase (pre-mitosis phase), increasing from 19.45 ± 1.91% in untreated cells to 39.75 ± 0.64% and 28.05 ± 3.04% in those treated with 2.5 µM and 25 µM acriflavine, respectively. Exposure to the highest dose also led to the detection of cells in the subG1 phase, with 16.15 ± 3.04%, which have a lower DNA content than cells in the G1 phase of the cell cycle, a phenomenon typically associated with dead or dying cells (Figure 5c). This suggests that acriflavine interferes with normal cell cycle progression, acting as an anti-mitotic agent and potentially contributing to its anti-proliferative effects.

Evaluation of the impact of HIF inactivation on hemangioblastomas was completed by assessing the type of cell death induced by acriflavine. To this end, hemangioblastoma cultures HB2, HB3, HB7, HB8, HB9, and HB10 were treated with acriflavine for 48 h, followed by propidium iodide and Annexin V-APC staining, and analyzed by flow cytometry. As shown in Figure 5d, acriflavine exposure led to a slight increase in both early (11.1 ± 0.57% and 10.35 ± 0.78%) and late apoptosis (12.85 ± 0.85% and 14.5 ± 0.71%), and a strong increase in dead cells positive for propidium iodide but negative for Annexin V (29.3 ± 0.3% and 44.3 ± 6.22%) at low and high doses compared to untreated primary HB cells, which displayed a proportion of early and late apoptotic cells and necrotic cells of 8.45 ± 1.06%, 8 ± 0.14%, and 11.55 ± 1.2%, respectively. These findings demonstrate that acriflavine exerts a potent cytotoxic and antiproliferative effect on primary HB cultures in a dose-dependent manner, leading to G2/M cell cycle arrest and the induction of necrosis.

## 4. Discussion

The standard treatment for hemangioblastomas associated with VHL disease remains surgical resection, despite the risk of neurological sequelae. However, due to the recurrent nature of VHL-related tumors and the risks associated with multiple surgeries, there is a rising interest in developing pharmacological treatments as a complementary strategy. Belzutifan (Welireg™), a HIF-2α inhibitor, is the only FDA-approved non-surgical option, but it has demonstrated only a 30% partial response rate in patients with CNS hemangioblastomas [42]. To explore alternative molecular targets as potential non-invasive therapeutic strategies, we established primary cell cultures from CNS hemangioblastomas of VHL patients undergoing surgery. Cellular and molecular analyses revealed overexpression of both HIF-α isoforms in HB cells, with HIF-1α levels slightly higher than HIF-2α. Interestingly, treatment of patient-derived HB cell culture with acriflavine led to a dose-dependent reduction in cell viability, associated with G2/M phase cell cycle arrest and enhanced necrosis. The mechanism underlying this HIF inhibitor-induced cytotoxicity could be attributed to the downregulation of HIF target genes, as the drug blocks dimerization with the β subunit and inhibits HIF transcriptional activity. The reduced expression of these target genes, which are involved in key cellular pathways such as survival, metabolism, glycolysis, proliferation, and angiogenesis, would ultimately lead to the death of HB cells. Our results therefore suggest that dual inhibition of HIF-1 and HIF-2 could represent a promising therapeutic approach for VHL-associated hemangioblastomas.

Despite the challenges inherent to the benign nature of these tumors, such as the slow proliferation rate of stromal, pericyte, and endothelial cells and the scarcity of published studies on primary HB cultures [52], we achieved successful establishment of primary HB cultures from 9 of 12 tumor samples, representing a 75% success rate. It is worth noting that, by determining the doubling time and life span, a strong correlation was identified between the proliferation patterns of patient-derived HB cell cultures and both the extent of hemangioblastoma burden and the incidence of tumor recurrence within a four-year period. These findings suggest that the use of primary cells represents a valuable tool, and that analyzing in vitro growth patterns may bring us a step closer to clinical translation for predicting aggressive proliferative behavior, which, in turn, could support the implementation of more intensive patient monitoring. However, further studies with larger cohorts are needed to validate these observations. Moreover, cell profiling of the proliferating primary HB cultures revealed a predominance of stromal cells, followed by pericytes, with a smaller proportion of endothelial cells. These results are in agreement with previous phenotypic characterizations of VHL-related hemangioblastomas [19,52,53], which identified three main cell types: stromal cells, the most abundant and which would be essential for tumor growth; pericytes, which would contribute to vascular stability; and endothelial cells, which, although less numerous, could play a crucial role in vessel formation in these highly vascularized tumors.

On the other hand, we observed that over half of the VHL patient-derived HB cells exhibited marked overexpression of both HIF-1α and HIF-2α. Notably, the remaining cells, despite showing low to moderate expression of HIF-α isoforms compared to the other HB cell cultures, still displayed levels elevated relative to those observed in glioblastoma cells, levels that can also be considered overexpression. These findings suggest a significant involvement of both hypoxia-inducible factors in the development and progression of CNS hemangioblastomas. These results are consistent with prior molecular and immunohistochemical research performed on tumor samples of CNS HB reporting increased HIF-α isoform levels [17,38] and elevated expression of HIF target genes like EGFR, TGF-α, and VEGF [54,55]. Notably, our results reveal that HIF-1α protein levels were higher than HIF-2α in primary HB cell cultures, a pattern that differs from that showed by renal cell carcinoma (RCC) carrying VHL-defective genes. In an in vitro model of VHL-null RCC, it has been found that transcriptional activity favors HIF-2α over HIF-1α, resulting in a distinct bias of alpha isoform expression toward HIF-2α [31]. Additionally, immunohistochemical analysis of nephrectomy specimens from VHL patients showed that HIF-1α expression is evident in the earliest detectable lesions, even in single tubular cells, whereas HIF-2α upregulation becomes significantly stronger in cystic and malignant lesions [56]. In this line, Kondo et al. [57] provided evidence that downregulation of HIF-2α using short hairpin RNAs was sufficient to suppress tumor formation in pVHL-defective renal carcinoma cells, reinforcing HIF-2α’s role as a major driver of VHL-associated RCC tumor development.

Currently, there is no effective pharmacological treatment capable of completely inhibiting HB growth. Therefore, as previously mentioned, surgical excision continues to be the treatment of choice for CNS hemangioblastomas. Although recent advancements in microsurgical techniques have significantly reduced mortality rates from 50% to 24%, the risk remains substantial as this procedure is performed in the core of the CNS on tumors with a strong tendency to bleed. To reduce the need for operative procedures and its associated complications, researchers emphasize the urgent need for noninvasive therapeutic alternatives effective in limiting tumor progression. Clinical trials involving patients with VHL disease and unresectable HBs have historically explored the use of tyrosine kinase inhibitors (targeting genes from the VEGFR, PDGFR, and EGFR families, which are regulated by the HIF pathway), such as semaxanib, vatalanib, sunitinib, and pazopanib, as well as monoclonal anti-VEGF antibodies (bevacizumab). However, the results have not been sufficiently robust to justify the widespread clinical application of these drugs [58,59,60,61]. Accordingly, over the past decade, HIF inhibition has become a major focus of research, as it is proposed that blocking these transcription factors would lead to the simultaneous downregulation of key proteins such as VEGF, PDGF, TGFA and GLUT, molecules essential for cell survival, growth, and metabolism, and whose gene expression is regulated by HIFs. In this context, previous findings, together with our results, which collectively reveal differential expression of HIF-1α and HIF-2α between CNS-HBs and RCC, suggest that therapeutic strategies targeting the HIF system should consider isoform-specific expression patterns in each VHL-associated tumor type. Therefore, considering the HIF-1α and HIF-2α overexpression distribution, the simultaneous inhibition of HIF-1α and HIF-2α may provide a more effective strategy for reducing the proliferative capacity of CNS-HBs compared to the use of selective HIF-2α inhibitors alone. This theory would explain the observed 30% partial response rate of Belzutifan in VHL-associated hemangioblastomas [42], while this drug and PT2385, a first-generation HIF-2α inhibitor [57], demonstrated clinical efficacy in VHL patients with renal cell carcinoma (RCC) [40,41,62].

Given the limited efficacy of selective HIF-2α inhibition in CNS hemangioblastomas, our study explores a broader therapeutic approach by targeting both HIF-1α and HIF-2α simultaneously. Unlike Belzutifan, which exclusively inhibits HIF-2α, acriflavine disrupts the dimerization of both HIF isoforms, effectively blocking their transcriptional activity. This mechanism is particularly relevant for VHL-associated hemangioblastomas, where both HIF-1α and HIF-2α are significantly overexpressed, as demonstrated in our primary patient-derived culture. Our group previously observed [51] that propranolol treatment reduced the expression of both HIF-1α and HIF-2α proteins in primary CNS-HB cultures, leading to cytotoxicity. However, the inhibitory activity of this drug on the alpha isoforms was not direct; rather, it exerts its effects indirectly by blocking β-adrenergic receptors, preventing cAMP signaling, and subsequently reducing PKA-mediated phosphorylation of Src, which ultimately affects the stabilization of HIF-1α and HIF-2α, lowering their levels. Therefore, it was not possible to confirm whether HIF-1/2 α inhibition was the main cause of cell death in HB cells, given that propranolol, a beta-adrenergic blocker, acts on several pathways. For this reason, in this study, we chose to use acriflavine, as it directly inhibits HIF and has been recognized as the most potent direct HIF inhibitor among the 336 FDA-approved drugs. In addition, it is a low-cost, water-soluble, and commercially available molecule. Originally used as an antiseptic agent a century ago [63], ACF has recently demonstrated efficacy against a broad spectrum of cancers, including glioblastoma, osteosarcoma, breast, lung, liver, colon, ovarian, and pancreatic cancers, as well as leukemia and melanoma [43,44]. It also exerts an impact on the immune system, promoting the infiltration of NK and CD8+ T cells into the tumor microenvironment [64]. Moreover, research in murine models of oxygen-induced ischemic retinopathy has shown that acriflavine, administered intraocularly or intraperitoneally, effectively inhibits retinal neovascularization and diminishes the expression of HIF-1-responsive genes in a dose-dependent manner [45]. Our results reveal that acriflavine treatment led to a significant reduction in cell viability in primary HB cultures, with a mean half-maximal inhibitory concentration (IC_50_) ranging from 2.5 to 5 μM. These values are comparable to the IC_50_ reported for the cytotoxic effects of acriflavine in glioblastoma cell lines, as well as in other tumor types such as lung and breast cancer cells [46,47,48]. Remarkably, acriflavine exhibited minimal cytotoxicity in healthy L929 cells, which were selected as a normal cell control model in accordance with the ISO 10993–5 standard for toxicity testing of Class II compounds [48]. Even at a high concentration of 10 μM, pure acriflavine reduced L929 cell viability by less than 30%. Compared with our findings in HB cultures and with results from other cancer cell types, this suggests that acriflavine exerts a more pronounced inhibitory effect on tumor cells than on non-cancerous cells [48]. On the other hand, flow cytometry analysis showed that HB cell death induced by this dual HIF-α inhibitor was mainly via necrosis, in contrast to that reported for glioblastomas, which underwent apoptosis as the main type of cell death after acriflavine incubation. This discrepancy in cell death pathways between hemangioblastoma and glioblastoma cells following acriflavine exposure may rely on the differential dependencies on HIF signaling for cell survival and proliferation. As demonstrated in this study, primary hemangioblastoma cultures produce higher levels of HIF-α isoforms than primary glioblastoma cells. Therefore, the abrupt disruption of these transcription factors by acriflavine may impair HIF-mediated survival and metabolic pathways, rapidly depleting the cell’s energy reserves. This energy crisis could trigger a cascade of events favoring necrosis, as apoptosis is an ATP-dependent process.

Hence, together these results suggest that dual inhibition of HIF-1α and HIF-2α could be a promising non-invasive strategy for the treatment of VHL-associated hemangioblastomas or as a complementary therapy to surgery. Moreover, the therapeutic use of dual HIF-α isoform inhibition in VHL-related tumors would have several clinical advantages. First, since HIF-1α and HIF-2α participate in interconnected signaling pathways, both proteins could compensate for each other in certain situations, such as selective inhibition of HIF-2α, which can trigger the activation of HIF-1α expression. Therefore, dual inhibition prevents this functional redundancy. Second, many tumors develop resistance to single inhibitors. In fact, in the VHL context, acquired resistance to HIF-2 antagonists has been reported in preclinical studies of PT2399 and in clinical studies of PT2385/MK-3475 (Phase 1) and PT2977 (Belzutifan) [40,65,66], underscoring the necessity of more comprehensive strategies to overcome resistance in HIF-2-targeted therapies. While Belzutifan selectively binds to the PAS-B domain of HIF-2α [67], exploiting subtle structural differences with the corresponding domain of HIF-1α, acriflavine interacts with common sites in both the PAS-A and PAS-B domains of both HIF isoforms, thereby preventing their dimerization with HIF-1β. Thus, dual targeting could reduce the risk of resistance by simultaneously disrupting HIF pathways. Moreover, the structure of acriflavine, a relatively small molecule, has allowed for the development of optimized formulations, enabling its encapsulation in various delivery systems to reduce in vivo acriflavine-induced toxicity and achieve sustained drug release for the treatment of multiple pathologies. Acriflavine was incorporated into poly-lactic-co-glycolic acid (PLGA)-formulated microparticles, which showed efficacy for long-term suppression of choroidal neovascularization [68]. Very recently, Korelidou et al. [48] reported the preparation and optimization of 3D-printed acriflavine-loaded reservoir-type implants, demonstrating antitumor effects against glioblastoma. Additionally, acriflavine is a low-cost, commercially available compound with a well-established safety profile, originally used as an antimicrobial agent [63]. These factors make it an attractive candidate for repurposing in oncology, particularly for rare diseases such as VHL. Finally, the availability of an alternative inhibitor to Belzutifan for treating CNS hemangioblastomas could facilitate the development of combination therapies, potentially exerting a synergistic effect in reducing tumor growth. This approach could enable a reduction in Belzutifan dosage, thereby reducing its most common adverse effects, such as decreased hemoglobin levels (with anemia occurring in 90% of patients, 7% of whom experience Grade 3 anemia), fatigue, increased creatinine, headache, dizziness, hyperglycemia, and nausea [59,62,69].

Several limitations should be considered in this study. Although primary hemangioblastoma cultures were successfully established from 9 out of 12 tumor samples obtained from patients diagnosed with VHL, a low-prevalence disease, the overall sample size remains limited. Future studies with larger cohorts of primary HB cell cultures are needed to validate the observed correlation between the proliferative rate of primary cultures and tumor-related characteristics. In this study, HIF target genes were not analyzed following acriflavine exposure due to the low yield of protein extracts; therefore, preclinical studies are necessary to demonstrate that the treatment induces downregulation of these genes, leading to growth suppression in patient-derived HB cell cultures. Similarly, larger cohorts may provide more detailed insights into the full spectrum of cellular targets of acriflavine and their contribution to treatment efficacy, as well as help investigate the selectivity of this HIF inhibitor for hemangioblastoma cells over non-tumor brain cells such as neurons and glia. Additionally, comparative studies between the HIF-2α selective inhibitor and acriflavine would provide valuable information. On the other hand, this study offers preclinical evidence supporting acriflavine’s therapeutic potential for VHL-associated hemangioblastomas, based on patient-derived cell cultures, which are better at preserving original tumor characteristics than established cell lines. However, they do not fully replicate the complex in vivo tumor microenvironment, potentially affecting treatment response. Thus, in vivo studies like patient-derived xenografts or genetically modified mouse models could provide more relevant acriflavine efficacy data, opening the possibility for the development of clinical trials to confirm these findings and evaluate acriflavine’s safety and efficacy in patients.

## 5. Conclusions

Our results reveal that the growth of patient-derived HB cultures is strongly associated with tumor burden and recurrence in patients with VHL, suggesting that the in vitro proliferation profile may mirror disease features related to CNS-HB progression. Furthermore, HIF-1α and HIF-2α were found to be overexpressed in the primary cells, and simultaneous inactivation of their transcriptional activity using acriflavine led to a significant reduction in cell viability, cell cycle arrest, and increased necrotic cell death. These findings therefore provide evidence that dual inhibition of HIF-1α and HIF-2α may represent a promising non-invasive therapeutic strategy for the management of VHL-associated hemangioblastomas.

## Figures and Tables

**Figure 1 biomedicines-13-01234-f001:**
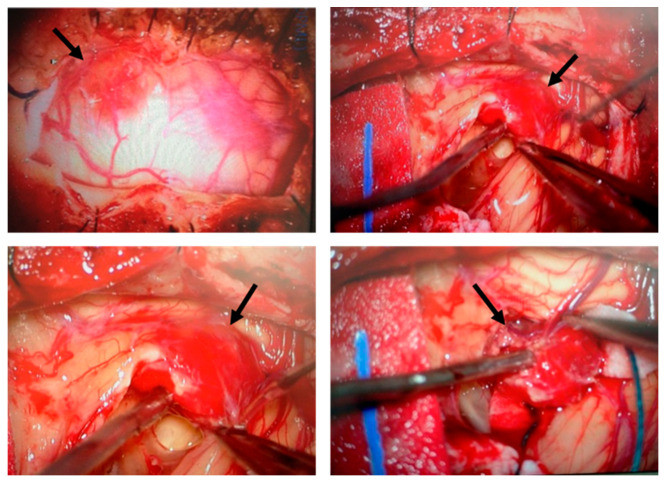
Surgical removal of a hemangioblastoma located in the cerebellum. Intraoperative images illustrate the surgical procedure for hemangioblastoma resection. The tumor (indicated by arrows), characterized by its highly vascularized nature, was carefully identified, isolated, and progressively dissected. This surgical approach is the standard treatment for CNS hemangioblastomas in patients with VHL disease, despite the risk of recurrence and the potential need for multiple interventions.

**Figure 2 biomedicines-13-01234-f002:**
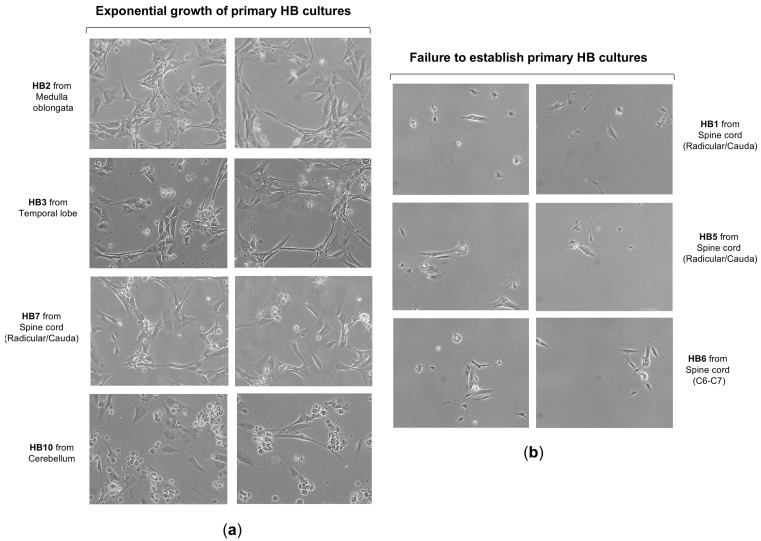
Growth and morphology of primary cell cultures from CNS hemangioblastomas. (**a**) Representative images show the successful establishment and expansion of primary cell cultures derived from CNS hemangioblastoma samples obtained from different anatomical locations, including the cerebellum, spinal cord, and temporal lobe. Primary HB cells predominantly exhibited a spindle-shaped or polygonal morphology and proliferated as monolayers under culture conditions. (**b**) A subset of HB tumor samples failed to develop proliferating primary cultures, particularly those derived from smaller tissue specimens.

**Figure 3 biomedicines-13-01234-f003:**
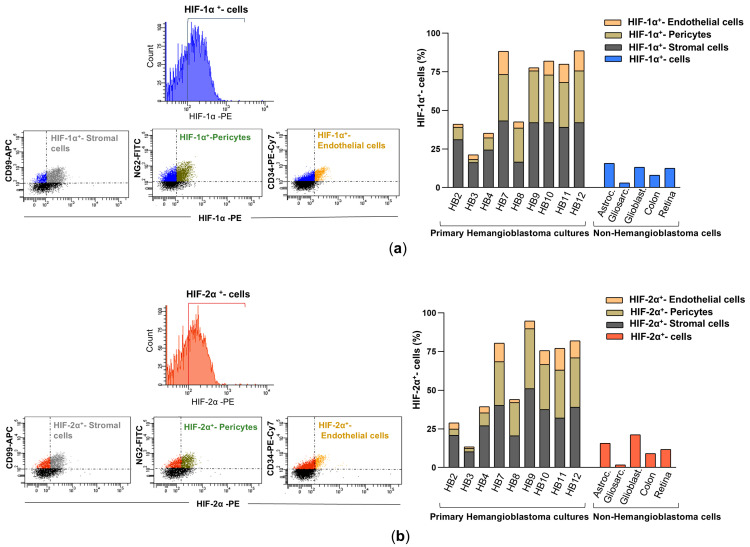
HIF-1α and HIF-2α expression in primary HB cells and their distribution among cellular components by flow cytometry analysis. (**a**) Representative fluorescence histogram (top left) for HIF-1α-PE single staining and dot plots (bottom left) for dual staining using fluorochrome-conjugated antibodies against HIF-1α-PE and cell-specific markers: CD99-APC for stromal cells, NG2-FITC for pericytes, and CD34-PE-Cy7 for endothelial cells. The histogram (right) depicts the percentage of HIF-1α-expressing cells within each cellular subpopulation: stromal cells (double-positive for HIF-1α-PE and CD99-APC), pericytes (double-positive for HIF-1α-PE and NG2-FITC), and endothelial cells (double-positive for HIF-1α-PE and CD34-PE-Cy7). These values are compared with HIF-1α expression levels in non-hemangioblastoma cells, including astrocytoma, gliosarcoma, glioblastoma, colorectal carcinoma, and human retinal pigment epithelium (ARPE) cell lines. (**b**) Illustrative fluorescence distribution chart (top left) for HIF-2α-PE staining, along with scatter plots (bottom left) depicting its co-detection with CD99-APC, NG2-FITC, or CD34-PE-Cy7. The histogram (right) shows the proportion of stromal (HIF-2α-PE^+^/CD99-APC^+^), pericytic (HIF-2α-PE^+^/NG2-FITC^+^), and endothelial (HIF-2α-PE^+^/CD34-PE-Cy7^+^) cells expressing HIF-2α, compared to non-hemangioblastoma cell lines positive for HIF-2α. Raw data for both histograms are available in the Appendix A (Data Set).

**Figure 4 biomedicines-13-01234-f004:**
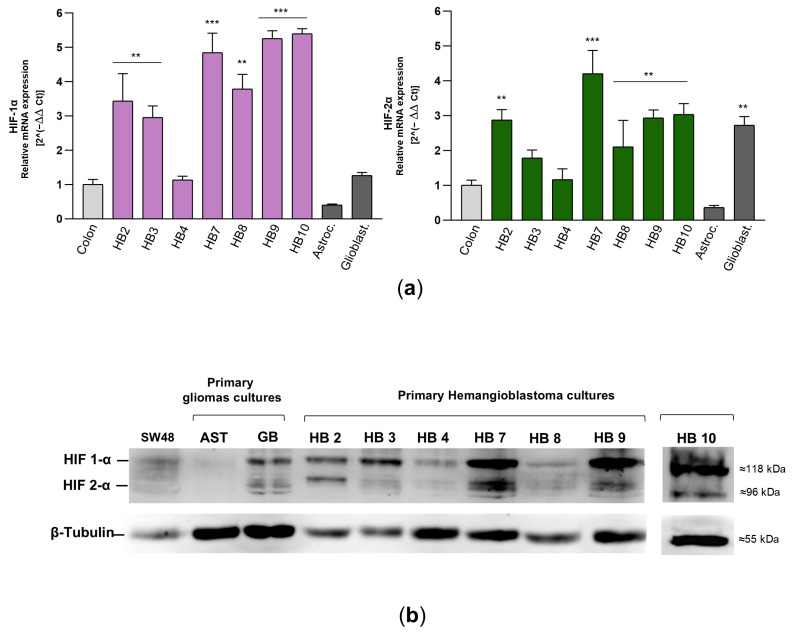
mRNA and protein levels of HIF-1α and HIF-2α in primary HB cells. (**a**) HIF-1α and HIF-2α transcript levels were determined by real-time RT-PCR in different primary cell cultures from hemangioblastomas and gliomas, as well as in the SW48 colorectal carcinoma cell line, which was used as a control. The mRNA levels of these factors were normalized to β-tubulin as an internal control and then to the corresponding HIF-α isoform mRNA from SW48 cells using the 2^−ΔΔCT^ method to determine relative gene expression with respect to the control cells. Results are presented as the mean ± SD from two independent experiments. Raw data for both histograms are available in the Appendix A (Data Set). Statistical significance was evaluated using Student’s *t*-test: ** *p* < 0.01, *** *p* < 0.001, compared to control colon cells. (**b**) Expression of HIF-1α and HIF-2α in hemangioblastoma (HB) cells, primary astrocytoma (AST) and glioblastoma (GB) cells, and SW48 cells was detected by Western blot analysis. In this figure, the grouped blots were cropped from different Western blot experiments, with β-tubulin as a loading control. The original blots are presented in the Appendix A.

**Figure 5 biomedicines-13-01234-f005:**
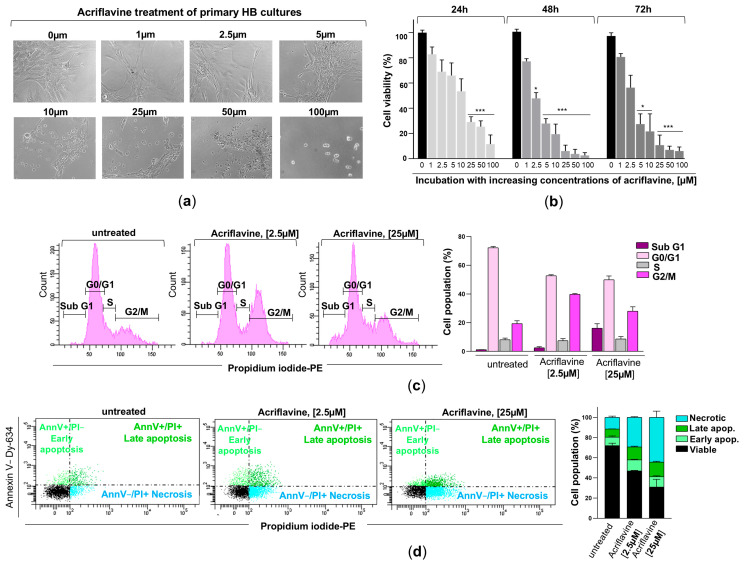
Effects of acriflavine exposure on cell viability, cell cycle, and cell death in primary HB cultures. (**a**) Representative phase-contrast microscopy images showing morphological changes in primary HB cells treated with increasing concentrations of acriflavine (1–100 µM) for 24 h. (**b**) Hemangioblastoma cell viability was assessed in cells incubated with (concentration indicated) or without acriflavine using the MTT assay. Cell viability was normalized to untreated cells for 24, 48, and 72 h and represented as a histogram (mean ± SD from two independent experiments). Statistical significance was assessed using Student’s *t*-test: * *p* < 0.05, *** *p* < 0.001, relative to HB cells not exposed to acriflavine. (**c**) Characteristic DNA content profiles from flow cytometry (left), displaying cells in sub-G1, G0/G1, S, and G2/M phases of the cell cycle, stained with propidium iodide after 24 h of incubation with 2.5 and 25 µM of acriflavine. The percentage of cells in each cell cycle phase is represented in histogram (right) from two separate experiments. Data are presented as means ± SD. (**d**) Representative dot plots (left) of flow cytometry analysis using Annexin V-Dy634 (AnnV) and propidium iodide (PI) staining, showing cell death by early (light green dots, single-positive for AnnV) and late (dark green dots, double-positive for AnnV and PI) apoptosis, and necrosis (blue dots, single-positive for PI) in primary HB cultures treated with two concentrations of acriflavine after 48 h. The histogram (right) shows the cell population undergoing necrotic, late, and early apoptotic cell death, as well as viable cells in untreated and treated cells (mean ± SD from two independent experiments). Raw data for histograms (**b**–**d**) can be found in the Appendix A (Data Set).

## Data Availability

The original contributions presented in this study are included in the article/Appendix A. Further inquiries can be directed to the corresponding authors.

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
