# Peer review of "Dual Inhibition of HIF-1α and HIF-2α as a Promising Treatment for VHL-Associated Hemangioblastomas: A Pilot Study Using Patient-Derived Primary Cell Cultures"

_biomedicines, 2025, doi:10.3390/biomedicines13051234_

Round 1

Reviewer 1 Report

Comments and Suggestions for Authors

The study of Perona-Moratalla et al. investigated the hemangioblastomas associated with the VHL-disease. They first created cell lines from patient-derived hemangioblastomas (HB), which were subsequently further characterized. Specifically, the authors focused on the expression of the HIF1alpha protein. They showed an increased expression in HB based on FACS; RT-qPCR and Western blotting experiments. Based on the results they tested the impact of acriflavine, an inhibitors that blocks the function of both HIF1-alpha und HIF2-alpha. They show that treating the cells with leads to reduced cell biability and altered cell cycle. Overall, this is a solid and interesting study that provides new insights into the molecular characteristics of VHL-associated hemangioblastomas and supports further investigation of HIF inhibitors as potential treatments. The work is well-executed and merits publication.

I have the following suggestion for improvement, which is however not absolutely necessary:

It would be nice to demonstrate that inhibition of HIF1 leads to altered gene expression of HIF target genes. If the hyphothesis of the authors is correct, one would expect a downregulation of these genes upon treatment. 

Author Response

Dear Reviewer 1, 

Please find attached the PDF file entitled " Response to review report (reviewer 1), biomedicines3560244,Perona-Moratalla et al., 2025 (pdf), in which we provide a point-by-point response to your comments and report.

Reviewer 2 Report

Comments and Suggestions for Authors

This study evaluates the anti-cancer effects of HIF-1α and HIF-2α inhibition on patient-derived VHL-related hemangioblastomas. The following suggestions aim to improve the manuscript.

  1. The subtitles of result section should be revised for more precision in reflecting the main findings.
  2. The authors should correlate cellular phonotypes, in terms of proliferation rate, doubling time, cell cycle progression, and HIF-1α and HIF-2α expression, with cancer severity and prognosis of the patients.
  3. The authors should evaluate the toxicological effects of acriflavine on cellular homeostasis in immortalized human neuronal cell lines.
  4. The mechanism underlying pharmaceutical inhibition of HIF-1α and HIF-2α on hemangioblastomas are unclear. The authors should provide experimental evidences or further discussion to address this.
  5. The Methods section requires revision. Specifically, catalog numbers for all experimental reagents, including antibodies, must be included in this section. Additionally, the cell number used in this study should be verified and revised as necessary.

Author Response

Dear Reviewer 2, 

Please find attached the PDF file entitled "Response to review report (Reviewer 2),biomedicines3560244,Perona-Moratalla et al, 2025", in which we provide a point-by-point response to your comments and report

Round 2

Reviewer 2 Report

Comments and Suggestions for Authors

The revised manuscript comprehensively addresses the principal concerns raised in the previous review. Substantial improvements in clarity, conciseness, and overall quality that make it suitable for publication in Biomedicines.